# Deconfined pseudocriticality in a model spin-1 quantum antiferromagnet

Vikas Vijigiri,[1] Sumiran Pujari,[1, *] and Nisheeta Desai[2]

[1]*Department of Physics, Indian Institute of Technology Bombay, Powai, Mumbai, MH 400076, India*
[2]*Department of Theoretical Physics, Tata Institute of Fundamental Research, Colaba, Mumbai, MH 400005, India*

Berry phase interference arguments that underlie the theory of deconfined quantum criticality (DQC) for $S = 1/2$ antiferromagnets have also been invoked to allow for continuous transitions in $S = 1$ magnets including a Néel to (columnar) valence bond solid (cVBS) transition. We provide a microscopic model realization of this transition on the square lattice consisting of Heisenberg exchange ($J_H$) and biquadratic exchange ($J_B$) that favor a Néel phase, and a designed $Q$-term ($Q_B$) interaction which favors a cVBS through large-scale quantum Monte Carlo (QMC) simulations. For $J_H = 0$, this model is equivalent to the $SU(3)$ $JQ$ model with a Néel-cVBS transition that has been argued to be DQC through QMC. Upon turning on $J_H$ which brings down the symmetry to $SU(2)$, we find multiple signatures – a single critical point, high quality collapse of correlation ratios and order parameters, "$U(1)$-symmetric" cVBS histograms and lack of double-peak in order parameter histograms for largest sizes studied near the critical point – that are highly suggestive of a continuous transition scenario. However, Binder analysis finds negative dips that grow sub-extensively that we interpret as these transitions rather being pseudocritical. This along with recent results on spin-$\frac{1}{2}$ models suggests that deconfined pseudocriticality is the more generic scenario.

The theory of deconfined quantum criticality (DQC) [1–4] has been of great interest as it lies beyond the Landau-Ginzburg-Wilson-Fisher paradigm. It posits a continuous transition between two symmetry unrelated phases – Néel and valence bond solid (VBS) – for spin-$\frac{1}{2}$ moments in $2 + 1d$. DQC is described as a gauge theory of fractionalized spinon degrees of freedom that deconfine only at the critical point. Their Higgs condensation leads to antiferromagnetic Néel order, while on the other side the confinement of the associated $U(1)$ gauge field [5–8] leads to the VBS. It has undergone a great deal of scrutiny [9] in various spin-1/2 models [10–21] and their SU($N$) generalizations [22–27]. Evidence for many features of DQC have been numerically seen in these studies, including classical loop models and dimer models in $3d$ [28–31], certain $1 + 1d$ spin-$\frac{1}{2}$ extensions [32–34], and fermionic models [35–39].However scaling violations have also been seen [40–46] that are still under debate. Not much is known though for $S > \frac{1}{2}$.

Our focus will be on spin-1 here. There are only a handful of works discussing possible DQC and none which have shown DQC behavior in microscopic model realizations. Previously, Ref. [47] argued for a possible DQC from a spin-nematic state to a VBS state based on field-theoretic arguments. Ref. [48] similarly conjectured a possible DQC from Néel to a bond-nematic or Haldane-nematic state which has been numerically investigated in Refs. [49–52]. It further conjectured possible DQC from Néel to cVBS as a "doubled" spin-$\frac{1}{2}$ DQC theory [53]. Briefly, the theory is formulated by taking two copies of an $SO(5)$ field theory with $k = 1$ Wess-Zumino-Witten term for a combined 5-component order parameter field made from the Néel and columnar-VBS fields [54, 55] that has been used to describe $S = \frac{1}{2}$ DQC. Upon invoking a strong ferromagnetic coupling between the two

copies to be consistent with spin-1 at low energies, it reduces to a "single" $SO(5)$ field theory now with a doubled Wess-Zumino-Witten term. It is the presence of this topological term which can allow for a DQC betwen Néel to cVBS (for details, see the supplementary of Ref. [48]).

We computationally investigate this latter scenario based on the following heuristic: $JQ$ models [10] have been crucial in probing DQC physics in large-scale simulations. The basic units are $SU(2)$ singlet projectors $P_{ij}^2 = \left(\frac{1}{4} - \mathbf{s}_i \cdot \mathbf{s}_j\right)$ for $S = \frac{1}{2}$ on bond $\langle ij \rangle$. One can extend [56] them to $SU(N)$ as $P_{ij}^N = \sum_{\alpha,\beta=1}^{N} |\alpha_i; \alpha_j\rangle\langle\beta_i; \beta_j|$. The $U(1)$-gauge fluctuations get suppressed as $N$ increases. This gives closer match between perturbation theory and numerical estimates of critical exponents in $N > 2$ microscopic models [57]. The $SU(3)$ $JQ$ model

$$H_{SU(3)} = -J_B \sum_{\langle ij \rangle} P_{ij}^3 - Q_B \sum_{\langle ijkl \rangle} \left(P_{ij}^3 P_{kl}^3 + P_{il}^3 P_{jk}^3\right) \quad (1)$$

can also be recast as a spin-1 model since $P_{ij} = \frac{1}{3}\{(\mathbf{S}_i \cdot \mathbf{S}_j)^2 - \mathbb{I}\}$ biquadratic exchange for $S = 1$ [22]. $\langle ijkl \rangle$ indexes elementary plaquettes of the square lattice with a clockwise indexing of the sites $i, j, k, l$. The first term favors Néel order. The $Q$-term favors a cVBS of spin-1 valence bonds (also $SU(3)$ singlets for Eq. 1). DQC behavior between Néel and VBS has been observed in the $SU(3)$ $JQ$ model up to system sizes $L = 48$ [23].

We now add $S = 1$ Heisenberg exchange favoring Néel order as an $SU(2)$-symmetric perturbation to study the same transition, i.e.

$$H_{SU(2)} = H_{SU(3)} + J_H \sum_{\langle ij \rangle} \{\mathbf{S}_i \cdot \mathbf{S}_j - \mathbb{I}\} \quad (2)$$

by varying $g \equiv \frac{Q_B}{J_B}$ for different $J_H$ using quantum Monte Carlo (QMC) methods [59–62]. We note for later discussion that our QMC method work with a ("doubled")

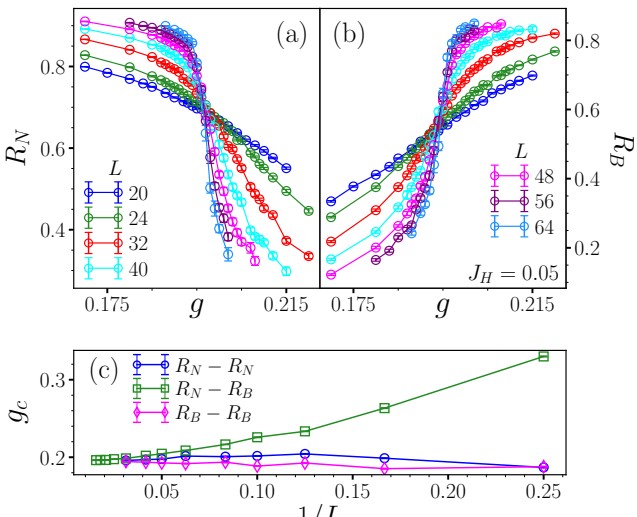

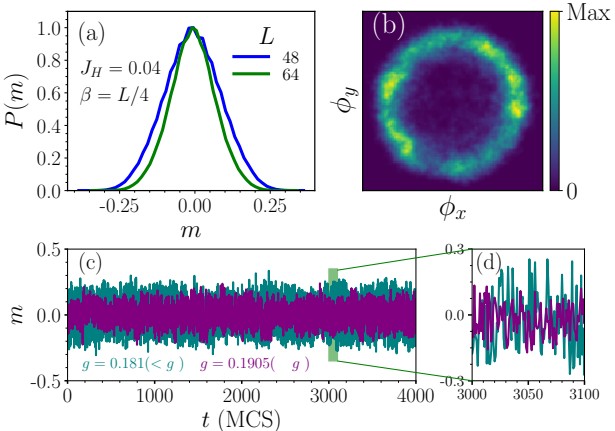

FIG. 1. Correlation ratios of (a) Néel and (b) cVBS order parameters ($R_N$ and $R_V$ [58]) $g = Q_B/J_B$, $J_H = 0.05$, $\beta = \frac{L}{4}$. (c) shows $g$ where the $R_N$ values cross for $L$ and $L/2$ versus $1/L$, similarly for $R_V$, and where $R_N$ and $R_V$ values cross for each $L$. The three estimates of $g_c(L)$ converge to $g_c \sim 0.195$ implying a direct transition without an intermediate phase.

FIG. 2. (a) Histogram of staggered magnetization $m$ (along $z$-direction) for $J_H = 0.04$ near $g_c \sim 0.19$. (b) Heat Map of the cVBS order parameters ($\phi_x, \phi_y$) shows "$U(1)$-symmetry" characteristic of DQC. (c) The time series plot for $m$ does not show any telegraphic switching behaviour ruling out a strong first-order transition. $L = 64$ in (b),(c).

Hilbert space of two "split $S = \frac{1}{2}$"s per site and a symmetrization step [63, 64] to restrict to the physical $S = 1$ subspace quite in analogy with doubled $S = \frac{1}{2}$ DQC theory. An earlier work [65] had studied the $J_B = 0$ case and found strong first-order behavior. We will thus focus on the vicinity of $H_{SU(3)}$, i.e. Eq. 2 where $i$ refers to sites, $\langle ij \rangle$ to nearest neighbor bonds, and $\mathbf{S}_i$ to spin-1 operators. We work in the units where $J_B = 1.0$. The following scenarios may be expected: (1) $SU(3)$ criticality becomes first-order right upon turning on $J_H$ and we see some cross-over physics for small values of the perturbation, (2) there is a regime of $J_H$ for which the doubled $S = 1/2$ DQC scenario obtains, or (3) there is weakly first-order or pseudocritical behaviour in this regime. For the second scenario, we expect to see stable exponents that are either $SU(3)$ exponents or a new set of $SU(2)$ exponents.

We probe the system by measuring intensive order parameters for the two phases. For the Néel phase, the staggered magnetization order parameter is $O_N \equiv \langle m^2 \rangle$, where $m = \frac{1}{N} \sum_{\mathbf{r}} e^{i(\pi,\pi) \cdot \mathbf{r}} S_{\mathbf{r}}^z$. For the columnar VBS phase, the cVBS order parameter is $O_V \equiv \langle (\phi_x^2 + \phi_y^2) \rangle$, where $\phi_\mu = \frac{1}{N} \sum_{\mathbf{r}} e^{i\pi \mathbf{e}_\mu \cdot \mathbf{r}} \{ (\mathbf{S}_{\mathbf{r}} \cdot \mathbf{S}_{\mathbf{r}+\mathbf{e}_\mu})^2 - \mathbb{I} \}/3$ [66]. The inverse temperature $\beta$ is set equal to $L/4$ to study ground state properties [67]. Fig 1 shows correlation ratios ($R$) of these order parameters [58] versus $g = \frac{Q_B}{J_B}$ for a representative value of $J_H = 0.05$. We see a clear crossing at the transition in Fig. 1a,b. The crossing points of both the Néel and VBS ratios converge versus $1/L$ (Fig. 1c) implying a direct transition between the two phases with

no intermediate phase.

We now look at order parameter histograms to probe the nature of this direct transition. No signature of two-peak behavior is seen near the transition as shown for staggered magnetization in Fig 2a. Telegraphic switching between the two order parameters is also absent near the transition (Fig 2c). This rules out the first scenario of a strongly first-order transition. Furthermore, "$U(1)$-symmetric" ($\phi_x, \phi_y$)-histograms are also seen near the transition for the largest system size studied (Fig 2b). In $S = \frac{1}{2}$ studies, this has been considered a key evidence of DQC. This is associated with the dangerous irrelevancy of the the operators in the DQC theory that capture the dominant quantum fluctuations out of the Néel phase. For $S = 1$, the appropriate operators are those of the doubled-DQC theory [68]. We therefore perform scaling collapses [69] of the order parameters and the correlation ratios as shown in Fig. 3. The scaling collapses are of high quality for all $J_H$ with the $\chi^2$ per degree of freedom being close to 1 throughout. Table I lists the exponents extracted from this finite-size scaling analysis. The exponents are stable to various protocols involving the range of the tuning parameter $g$ and system sizes used for the collapses.

The first thing of note is that the anomalous exponents $\eta_N$ and $\eta_V$ are markedly different as soon as $J_H \neq 0$. For $J_H = 0$, we obtain $SU(3)$ exponents in overall agreement with earlier work [23] though our best estimate for the Néel correlation exponent $\nu_N$ is different. The collapse quality when $\nu_N$ is set same as $\nu_V$ ($\sim 0.63$) is not significantly worse. The equality $\nu_N = \nu_V$ is expected in the theory of DQC. This marked difference of $\eta_N, \eta_V$ from $J_H = 0$ suggests that $SU(3)$ criticality is not obtained when $J_H \neq 0$, i.e. the $SU(2)$ perturbation changes the

TABLE I. Critical exponents ($\eta_V, \eta_N, \nu$) as obtained from scaling collapse analysis of the order parameters ($O_{N,B}$) for different $J_H$. $L = 24, 32, 40, 48, 64$ used here. Additional corroborating analysis with correlation ratios is shown in the Ref. [70].

| $J_H$ | $\nu_N$ | $\nu_V$ | $\eta_N$ | $\eta_V$ | $g_{cN}$ | $g_{cV}$ | $\chi_N^2$ | $\chi_V^2$ |
|---|---|---|---|---|---|---|---|---|
| 0.0 | 0.53(3) | 0.63(1) | 0.44(5) | 0.49(2) | 0.168(1) | 0.167(1) | 1.08-1.68 | 1.69-2.46 |
| 0.01 | 0.45(2) | 0.54(3) | 0.23(3) | 0.42(4) | 0.174(1) | 0.171(1) | 1.19-1.63 | 1.38-1.73 |
| 0.025 | 0.43(3) | 0.46(4) | 0.15(9) | 0.38(2) | 0.182(1) | 0.180(1) | 0.75-1.46 | 0.8-1.4 |
| 0.04 | 0.40(2) | 0.43(5) | 0.13(7) | 0.30(8) | 0.19(1) | 0.189(1) | 1.06-1.67 | 1.09-1.5 |
| 0.05 | 0.39(4) | 0.38(5) | 0.20(9) | 0.29(6) | 0.196(1) | 0.195(1) | 0.87-1.31 | 0.87-1.96 |
| 0.07 | 0.38(2) | 0.39(3) | 0.10(4) | 0.10(4) | 0.207(1) | 0.206(1) | 1.52-2.54 | 1.04-1.77 |
| 0.1 | 0.35(4) | 0.35(3) | -0.03(5) | -0.03(2) | 0.224(1) | 0.224(1) | 1.24-3.28 | 0.99-1.97 |
| 0.15 | 0.33(2) | 0.33(1) | 0.00(8) | -0.12(8) | 0.253(1) | 0.253(1) | 1.42-1.79 | 1.15-1.63 |

universality class. This is not entirely unexpected and can be taken as evidence for the doubled spin-1/2 DQC scenario [48]. However, there is a slow drift in the exponents as $J_H$ increases which argues against a stable set of exponents [71] as expected in the second scenario. The drift is noticeable even within the accuracy levels achieved by us. This level of accuracy in the estimation of critical exponents is not unusual in numerical studies of DQC. Similarly, the best estimates of $\nu_N$ and $\nu_V$ for each value of $J_H$ do not match in all cases, but again setting them equal does not lead to significant loss of collapse quality. Nevertheless, we certainly see that the anomalous exponents $\eta_N$, $\eta_V$ are not small. This is one of the expectations in the theory of DQC which is strikingly different than conventional second-order critical points, and seen in previous $S = \frac{1}{2}$ studies. Eventually for large enough $J_H = 0.1$ and $0.15$, the anomalous exponents go negative indicating first-order behavior. The correlation exponents $\nu_N$, $\nu_V \sim \frac{1}{2+1}$ for these $J_H$ values as well. This is to be expected when $J_H$ becomes large enough since first-order behaviour has been seen for $J_B = 0$ [65].

From the preceding discussions, the Néel-cVBS transition in Eq. 2 appears continuous up to $J_H \sim 0.07$. However, due to the observed drifts in the exponents, we further examine the Binder ratios of the magnetization order parameter. We would expect them to behave similar to the correlation ratios. Fig 4 shows this ratio, defined as $\frac{5}{2}(3 - \frac{\langle m^4 \rangle}{\langle m^2 \rangle^2})$, with a clearly visible crossing at the transition point. Equally noteworthy is the clear dip below zero near the transition. This is seen for all values of $J_H$ [70]. A characteristic of first order transitions is that this dip grows extensively with system size [72]. In Ref [42], where the $J_H = 0$ case was studied, it had been noted that this dip grows sub-extensively with system size and interpreted as evidence for a continuous transition. We similarly find sub-extensively growing dips for small $J_H$ as shown in Fig 5. As $J_H$ grows larger, it eventually grows as $L^2$ as expected for a first order transition [73] concomitantly with $\nu_N$, $\nu_V \sim \frac{1}{2+1}$.

Given the drifting exponents seen earlier that argues against a bonafide DQC scenario [74], we rather interpret the sub-extensive growing Binder dips along with the

$U(1)$-symmetric VBS histograms as evidence for deconfined pseudocriticality. In other words, the scenario of a doubled $S = \frac{1}{2}$ DQC provides a framework to understand the above set of numerical results, but the deconfinement gets curtailed at much larger length scales (dependent on $J_H$) than the lattice scale leading to the observed pseudocritical behavior. In terms of QMC simulations, we imagine it as pseudo-DQC ensembles occurring in both the split $S = \frac{1}{2}$ Hilbert spaces in (QMC) space-time which then gets "inherited" by the $S = 1$ system under projection back to the fully symmetric subspace. This also implies a revision of the earlier interpretation of DQC in

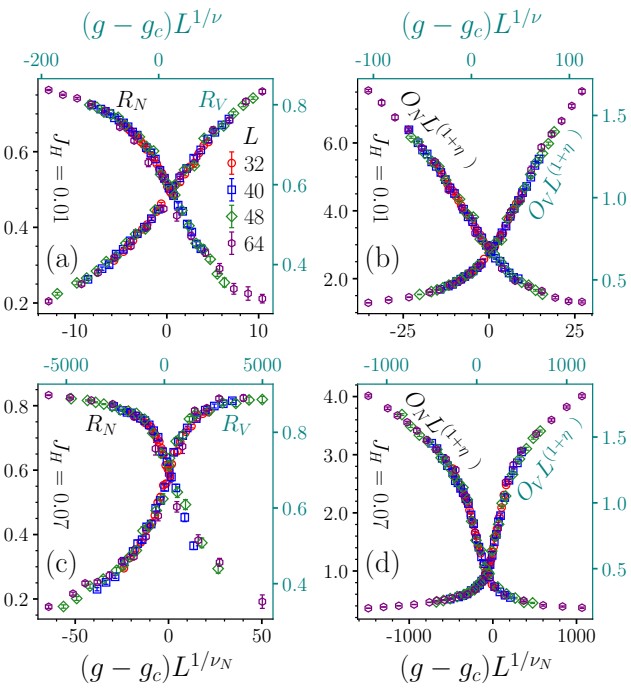

FIG. 3. Scaling collapse of correlation ratios ($R_N$, $R_V$) and order parameters ($O_N$, $O_V$) for $J_H = 0.01$ and $0.07$ with best estimates of the critical exponents (Table I). ($\beta = \frac{L}{4}$). Note the high quality of collapse. Also associated $\chi^2$ values in Table I. This highly suggests a continuous transition within the doubled-DQC framework.

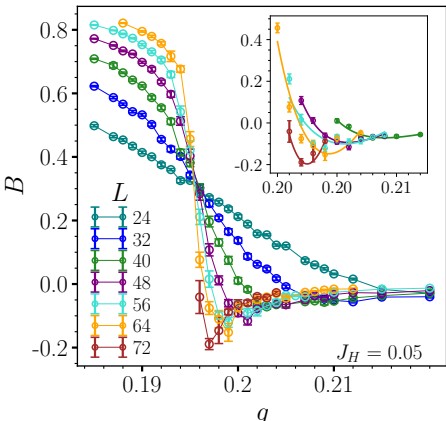

FIG. 4. Binder ratio plotted versus fugacity $g = Q_B/J_B$ for $J_H = 0.05$ ($\beta = \frac{L}{4}$). Similar plots with negative dips are obtained for other $J_H$ values [70]. This calls into question the continuous nature of the transition. The inset shows interpolations done in the vicinity of the Binder dip to extract an estimate of the dip magnitude for Fig. 5.

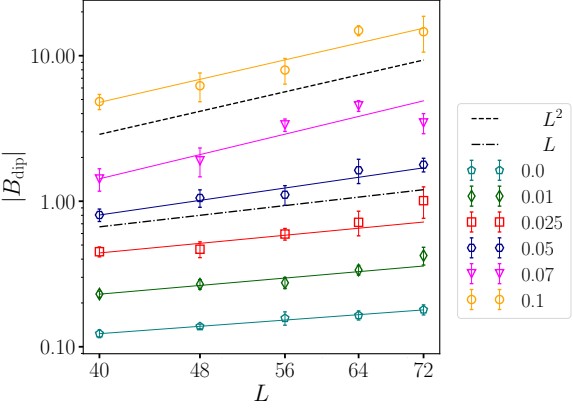

FIG. 5. (Negative) Binder dip magnitude versus $L$ for different $J_H$ shown on log-log scale. It scales subextensively for small $J_H$. We interpret this as (deconfined) pseudocriticality. The dashed lines are $L$ and (extensive) $L^2$ power laws. The different $J_H$ curves have been scaled by different constant factors for clarity.

$SU(3)$ $JQ$ model [42].

We situate our interpretation of deconfined pseudocriticality in our $S = 1$ microscopic model in the light of recent developments that have thrown open the issue of the second-order nature of DQCs [75–78]. These arose in the context of earlier works regarding emergent symmetry expectations at these transitions when interpreted in a $SO(5)$ framework with a combined order parameter built out of the Néel and VBS order parameters [54, 55]. This would imply an enhanced emergent symmetry between the two symmetry unrelated order parameters. This basic expectation has been numerically studied in various cases [46, 79, 80], in certain unconventional transitions [81] including in one dimension [82, 83] and classical dimer models [84]. System size restrictions however can make interpretation of emergent symmetry tricky. From the theory side, conformal bootstrap results [85] pointed out strong constraints on the scaling dimensions that apparently rule out emergent symmetry in $2 + 1d$ DQC. The notion of DQC as pseudocriticality was thus conjectured [86, 87] in order to reconcile with the conformal bootstrap result. The idea is that the renormalization group (RG) fixed point for DQC does not reside in the (physical) $2 + 1d$, but slightly below it (see Fig. 1 of Ref. [86]) such that pseudocritically slow flows obtain near the critical point $g_c$. For more discussion on these issues, see this recent review [88]. Such slow RG flows may account for the good scaling collapse seen in numerical data for accessible system sizes *along* with the observed drifts in the critical exponents. Recent quantum entanglement based results [76–78] have taken forward a similar line of argument for $SU(N)$ models in general, however another work [89] provides a counterargument.

It is noteworthy that even before the emergent symmetry point of view gained currency, numerical studies of DQC had seen anomalous scaling corrections whose origin was unclear. Ref. 44 gave an explanation based on scaling corrections inherent in the effective $U(1)$ gauge theory of the deconfined spinons. Another explanation based on two different length scales associated with spinon correlations and VBS domain wall sizes diverging with different exponents has also been proposed [45]. In the context of our interpretation, pseudocriticality indicates at the confinement of the putative spinons of the $U(1)$ gauge theory framework for all values of the tuning parameter as mentioned previously. Near the transition, the confinement length scale must become very large ($L \gtrsim O(100)$ given the high quality of scaling collapses seen) compared to the lattice scale [90] but remain finite due to pseudocritical nature of the RG flows. The present consensus seems to be veering towards deconfined pseudocriticality based on recent $S = \frac{1}{2}$ results, and our work provides a microscopic $S = 1$ model for this scenario. This opens a question in the context of $SU(N)$ DQC. On one hand, theoretical expectations based on the suppression of gauge fluctuations with increasing $N$ make the case for bonafide DQC. On the other hand, no negative Binder dips has been numerically seen in $SU(2)$ $JQ$ models for largest sizes studied. This makes the pattern of Binder dip growth with respect to $N$ mysteriously non-monotonic [91]. It will be an useful theoretical advance and strong evidence for the pseudocriticality scenario if the sub-extensive growth of negative Binder ratio dips can be linked to the pseudocritcal RG flows of Refs. [86, 87].

*Acknowledgements:–* We acknowledge useful discussions with Fabien Alet, Subhro Bhattacharjee, Kedar Damle, Prashant Kumar and Adam Nahum. VVi was

supported by the institute post-doctoral fellowship program at IIT Bombay, and in part by the International Centre for Theoretical Sciences (ICTS) during the program - 8th Indian Statistical Physics Community Meeting (code: ICTS/ISPCM2023/02). SP acknowledges funding support from SERB-DST, India via Grants No. SRG/2019/001419 and MTR/2022/000386. Partial support by Grant No. CRG/2021/003024 is also acknowledged. ND was initially supported by National Postdoctoral Fellowship of SERB, DST, Govt. of India (PDF/2020/001658) at the department of Theoretical Physics, TIFR and presently by the TIFR postdoctoral fellowship. The numerical results were obtained using the computational facilities of the Department of Physics, Indian Institute of Technology (IIT) Bombay.

---

\* sumiran.pujari@iitb.ac.in

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

# Supplementary information for "Deconfined pseudocriticality in a model spin-1 quantum antiferromagnet"

Vikas Vijigiri,[1, *] Nisheeta Desai,[2] and Sumiran Pujari[1, †]

[1]*Department of Physics, Indian Institute of Technology Bombay, Powai, Mumbai, 400076, India*

[2]*Department of Theoretical Physics, Tata Institute of Fundamental Research, Mumbai, 400005, India*

(Dated: June 14, 2024)

Here we present additional data and plots to document the data sets collected during this work for the various Heisenberg exchange coupling $(J_H)$ values studied in the main text.

## CONTENTS

## I. MODEL AND OBSERVABLES

We recapitulate again the various definitions related to the model and observables introduced in the main text for convenience. The $S = 1$ Hamiltonian (Eq. 1 and Eq. 2 of the main text) is given by,

$$
\begin{aligned}
H = {} & J_H \sum_{\langle ij \rangle} \{\mathbf{S}_i \cdot \mathbf{S}_j - \mathbb{I}\} - \frac{J_B}{3} \sum_{\langle ij \rangle} \{(\mathbf{S}_i \cdot \mathbf{S}_j)^2 - \mathbb{I}\} \\
& - \frac{Q_B}{9} \sum_{\langle ijkl \rangle} \left[ \{(\mathbf{S}_i \cdot \mathbf{S}_j)^2 - \mathbb{I}\}\{(\mathbf{S}_k \cdot \mathbf{S}_l)^2 - \mathbb{I}\} + \{(\mathbf{S}_i \cdot \mathbf{S}_l)^2 - \mathbb{I}\}\{(\mathbf{S}_j \cdot \mathbf{S}_k)^2 - \mathbb{I}\} \right]
\end{aligned}
\tag{1}
$$

where $J_H, J_B$ are the coefficients of the Heisenberg and Biquadratic exchange terms, and $Q_B$ is the coefficient of the designer $Q$-term composed of biquadratic exchanges respectively.

---

* vikas.v@iitb.ac.in

† sumiran@phy.iitb.ac.in

The following structure factors

$$\mathcal{A}(\mathbf{q}) = \frac{1}{N^2} \sum_{\mathbf{r},\mathbf{r}'} e^{i(\mathbf{r}-\mathbf{r}')\cdot\mathbf{q}} \langle S_{\mathbf{r}}^z S_{\mathbf{r}'}^z \rangle \tag{2}$$

$$\mathcal{B}(\mathbf{q}) = \frac{1}{N^2} \sum_{\mathbf{r},\mathbf{r}'} e^{i(\mathbf{r}-\mathbf{r}')\cdot\mathbf{q}} \langle (\mathbf{S}_{\mathbf{r}} \cdot \mathbf{S}_{\mathbf{r}'} - \mathbb{I}) \rangle \tag{3}$$

$$\mathcal{C}^{\mu}(\mathbf{q}) = \frac{1}{N^2} \sum_{\mathbf{r},\mathbf{r}'} e^{i(\mathbf{r}-\mathbf{r}')\cdot\mathbf{q}} \langle (\mathbf{S}_{\mathbf{r}} \cdot \mathbf{S}_{\mathbf{r}+\mathbf{e}_{\mu}} - \mathbb{I}) \times (\mathbf{S}_{\mathbf{r}'} \cdot \mathbf{S}_{\mathbf{r}'+\mathbf{e}_{\mu}} - \mathbb{I}) \rangle \tag{4}$$

$$\mathcal{D}^{\mu}(\mathbf{q}) = \frac{1}{9N^2} \sum_{\mathbf{r},\mathbf{r}'} e^{i(\mathbf{r}-\mathbf{r}')\cdot\mathbf{q}} \langle ((\mathbf{S}_{\mathbf{r}} \cdot \mathbf{S}_{\mathbf{r}+\mathbf{e}_{\mu}})^2 - \mathbb{I}) \times ((\mathbf{S}_{\mathbf{r}'} \cdot \mathbf{S}_{\mathbf{r}'+\mathbf{e}_{\mu}})^2 - \mathbb{I}) \rangle \tag{5}$$

made out of real-space correlators can serve as order parameters for Néel and VBS ordering. As a side remark, $\mathcal{B}(\mathbf{q})$ can be measured during loop update in SSE. However, we do not need to do this as $\mathcal{A}(\mathbf{q})$ which also detects Néel ordering can measured in a much simpler way. Furthermore, in presence of $SU(2)$ symmetry they are linearly related to each other. Therefore, we focused on $\mathcal{A}(\mathbf{q})$ to probe Néel order at the antiferromagnetic ordering wavevector on the square lattice $((\pi,\pi)$ when lattice constants are set to unity) as mentioned in the main text as well. The corresponding Néel correlation ratio $R_N$ is defined to be

$$R_N \equiv \frac{R_x + R_y}{2} \tag{6}$$

where,

$$R_x \equiv 1 - \frac{\mathcal{A}(\pi + \frac{2\pi}{L}, \pi)}{\mathcal{A}(\pi,\pi)} \tag{7}$$

$$R_y \equiv 1 - \frac{\mathcal{A}(\pi, \pi + \frac{2\pi}{L})}{\mathcal{A}(\pi,\pi)} \tag{8}$$

Similarly, both $\mathcal{C}(\mathbf{q})$ and $\mathcal{D}(\mathbf{q})$ made out of bond-bond correlators of Heisenberg and biquadratic couplings can be used to probe VBS order at its ordering wavevector on the square lattice $((\pi,0)$ and $(0,\pi)$ when lattice constants are set to unity). Either one would thus suffice for the study of Néel-VBS transitions. The Heisenberg bond energy based VBS correlation ratio $R_{V'}$ is defined to be

$$R_{V'} \equiv \frac{R_{V'_x} + R_{V'_y}}{2} \tag{9}$$

where,

$$R_{V'_x} \equiv 1 - \frac{\mathcal{C}(\pi + \frac{2\pi}{L}, 0)}{\mathcal{C}(\pi,0)} \tag{10}$$

$$R_{V'_y} \equiv 1 - \frac{\mathcal{C}(0, \pi + \frac{2\pi}{L})}{\mathcal{C}(0,\pi)} \tag{11}$$

We have rather focused on the biquadratic bond energy based VBS observables in the paper due to better statistics while estimating it during QMC compared to the Heisenberg bond energy based observables as discussed also in a footnote in the main text. The corresponding VBS correlation ratio is thus defined to

$$R_V \equiv \frac{R_{V_x} + R_{V_y}}{2} \tag{12}$$

where,

$$R_{V_x} = 1 - \frac{\mathcal{D}(\pi + \frac{2\pi}{L}, 0)}{\mathcal{D}(\pi,0)} \tag{13}$$

$$R_{V_y} = 1 - \frac{\mathcal{D}(0, \pi + \frac{2\pi}{L})}{\mathcal{D}(0,\pi)} \tag{14}$$

Finally, from the above we can also see the correspondence between the notation of the main text and that used here for the order parameters as

$$O_N = \mathcal{A}(\pi,\pi) \tag{15}$$
$$O_V = \mathcal{D}^x(\pi,0) + \mathcal{D}^y(0,\pi) \tag{16}$$

## II.   DATA SETS

### A.   Convergence with inverse temperature $\beta$

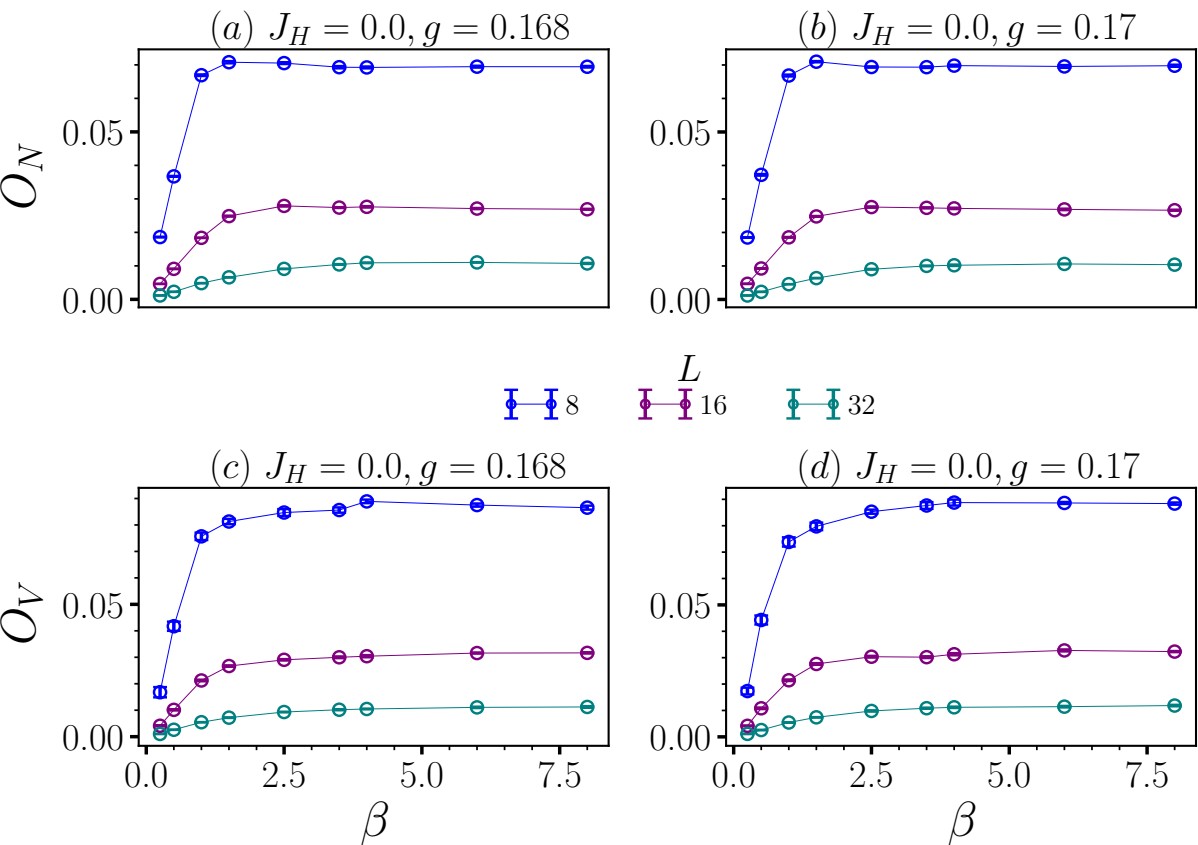

FIG. 1. Néel and cVBS order parameters ($O_N$ and $O_V$) versus the inverse temperature $\beta = \frac{1}{T}$. $\frac{J_H}{J_B} = 0.0$, $g \equiv \frac{Q_B}{J_B} \sim 0.17$. We recall that $J_B$ is set to 1 everywhere. Similar behaviour is present everywhere in the parameter regimes explored as illustrated through the next two figures. We see that $\beta = \frac{L}{4}$ is more than sufficient for making conclusions about ground state properties.

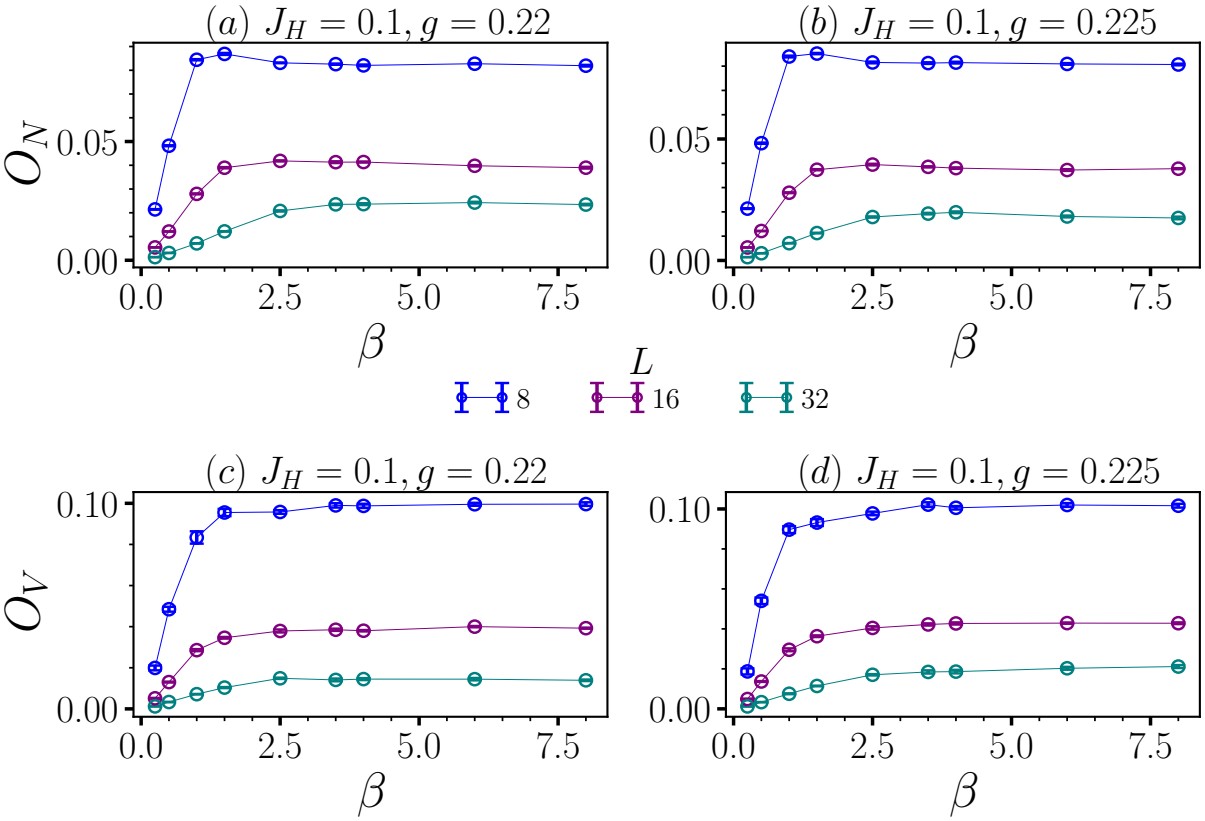

FIG. 2. Néel and cVBS order parameters ($O_N$ and $O_V$) versus the inverse temperature $\beta = \frac{1}{T}$. $\frac{J_H}{J_B} = 0.1$, $g \equiv \frac{Q_B}{J_B} \sim 0.22$.

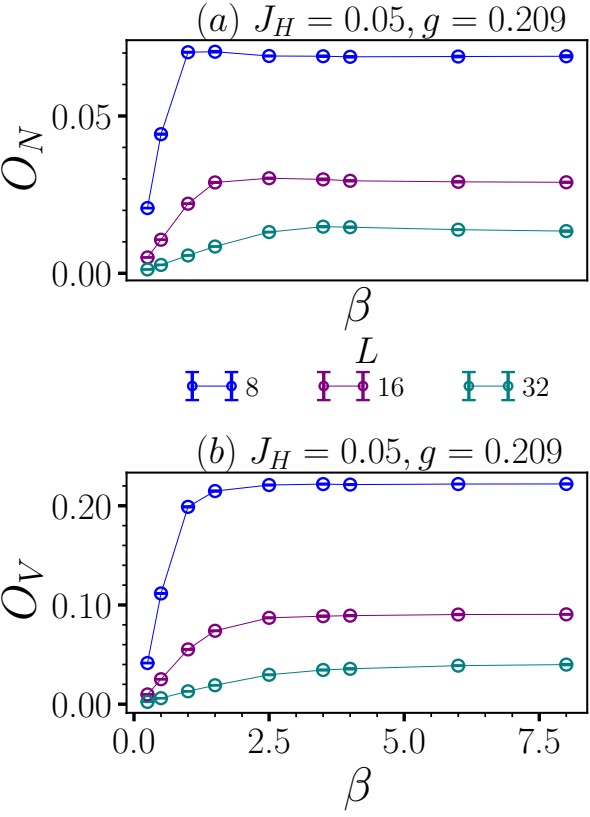

FIG. 3. Néel and cVBS order parameters ($O_N$ and $O_V$) versus the inverse temperature $\beta = \frac{1}{T}$. $\frac{J_H}{J_B} = 0.0$, $g \equiv \frac{Q_B}{J_B} \sim 0.21$.

## B.  Correlation Ratios

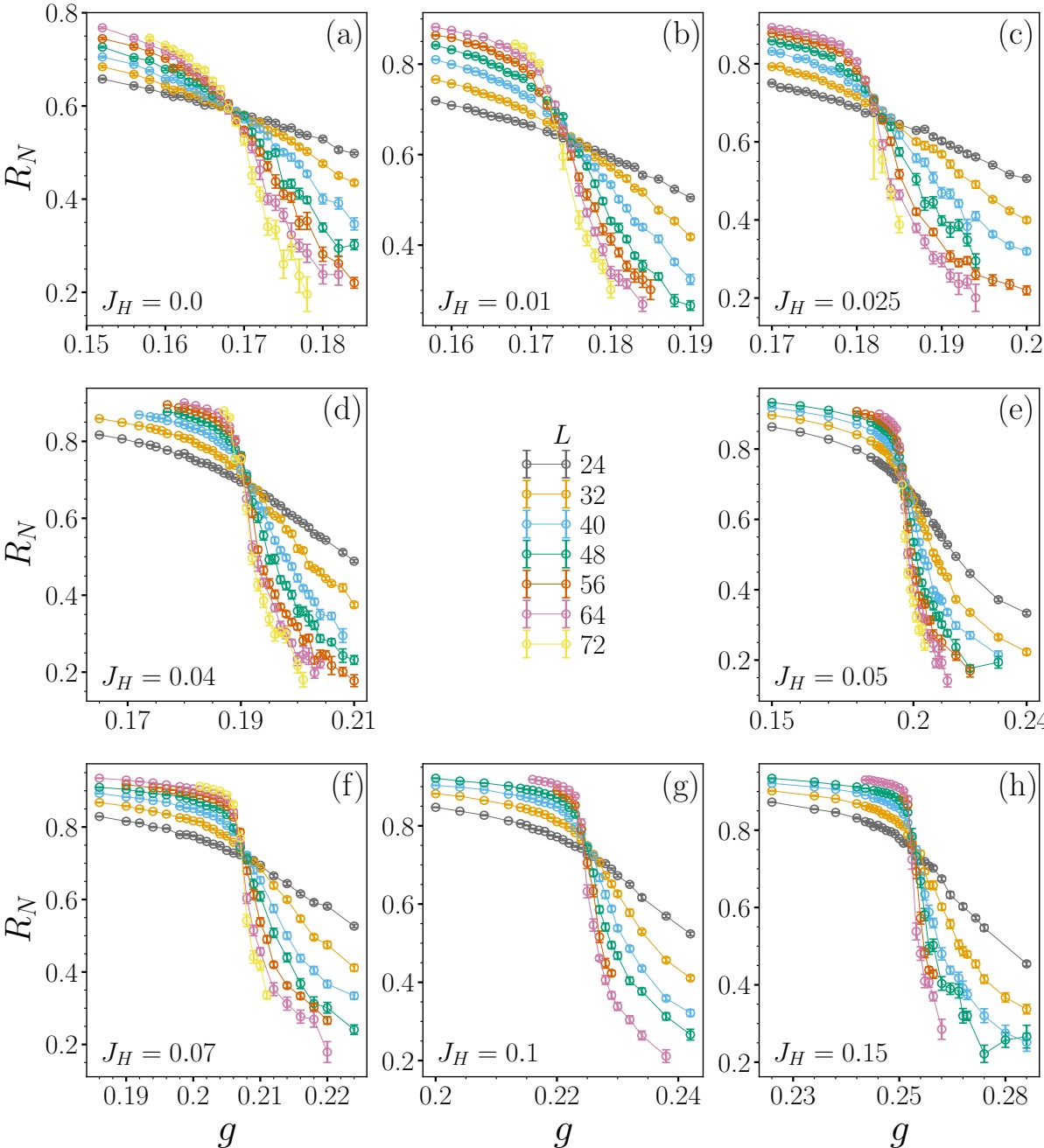

FIG. 4.  This figure expands on Fig. 1a of the main text to document all the $R_N$ correlation ratio data sets collected. $\beta = \frac{L}{4}$ throughout.

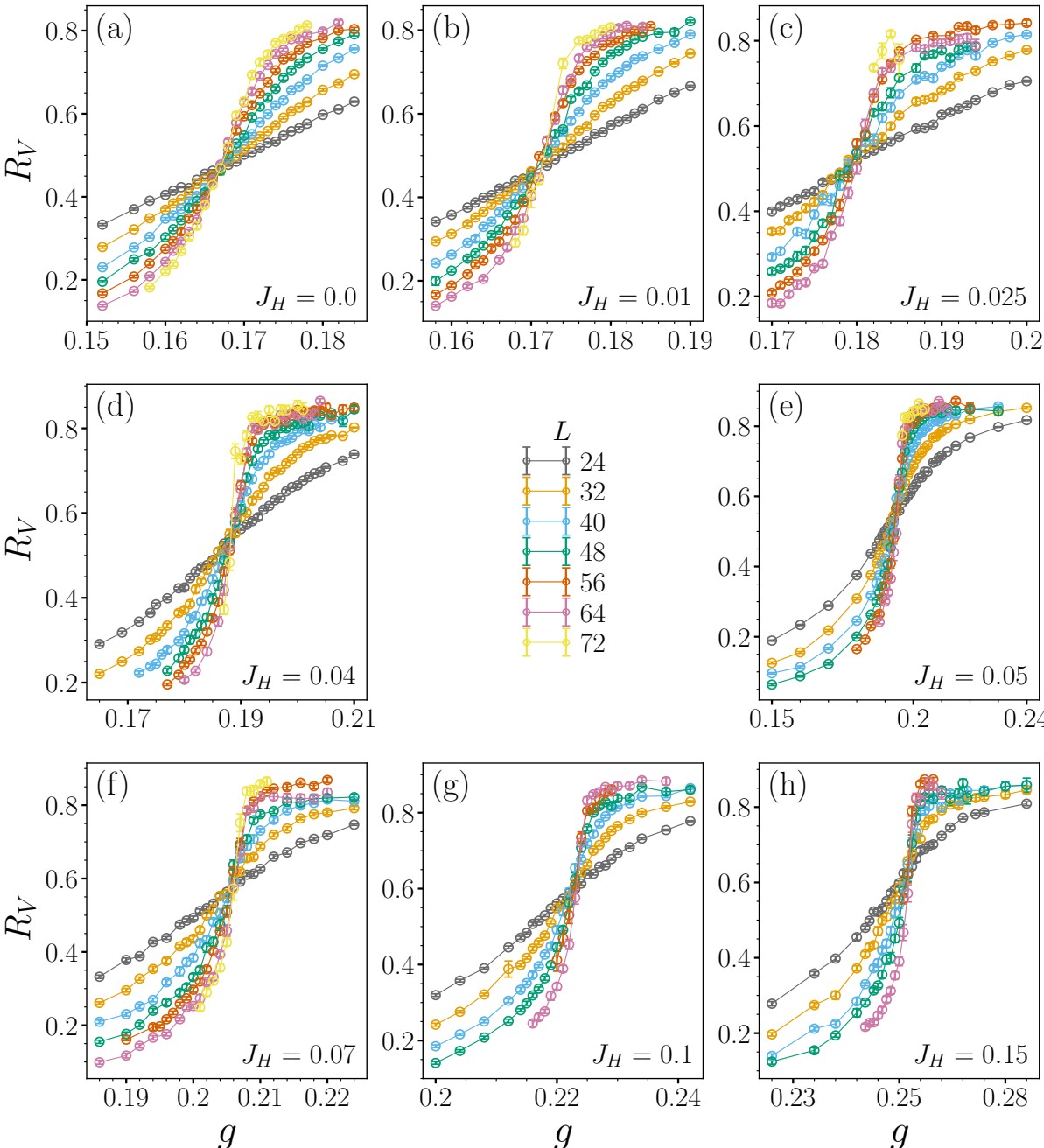

FIG. 5. This figure expands on Fig. 1b of the main text to document all the $R_V$ correlation ratio data sets collected. $\beta = \frac{L}{4}$ throughout.

## C.  Staggered magnetization histograms

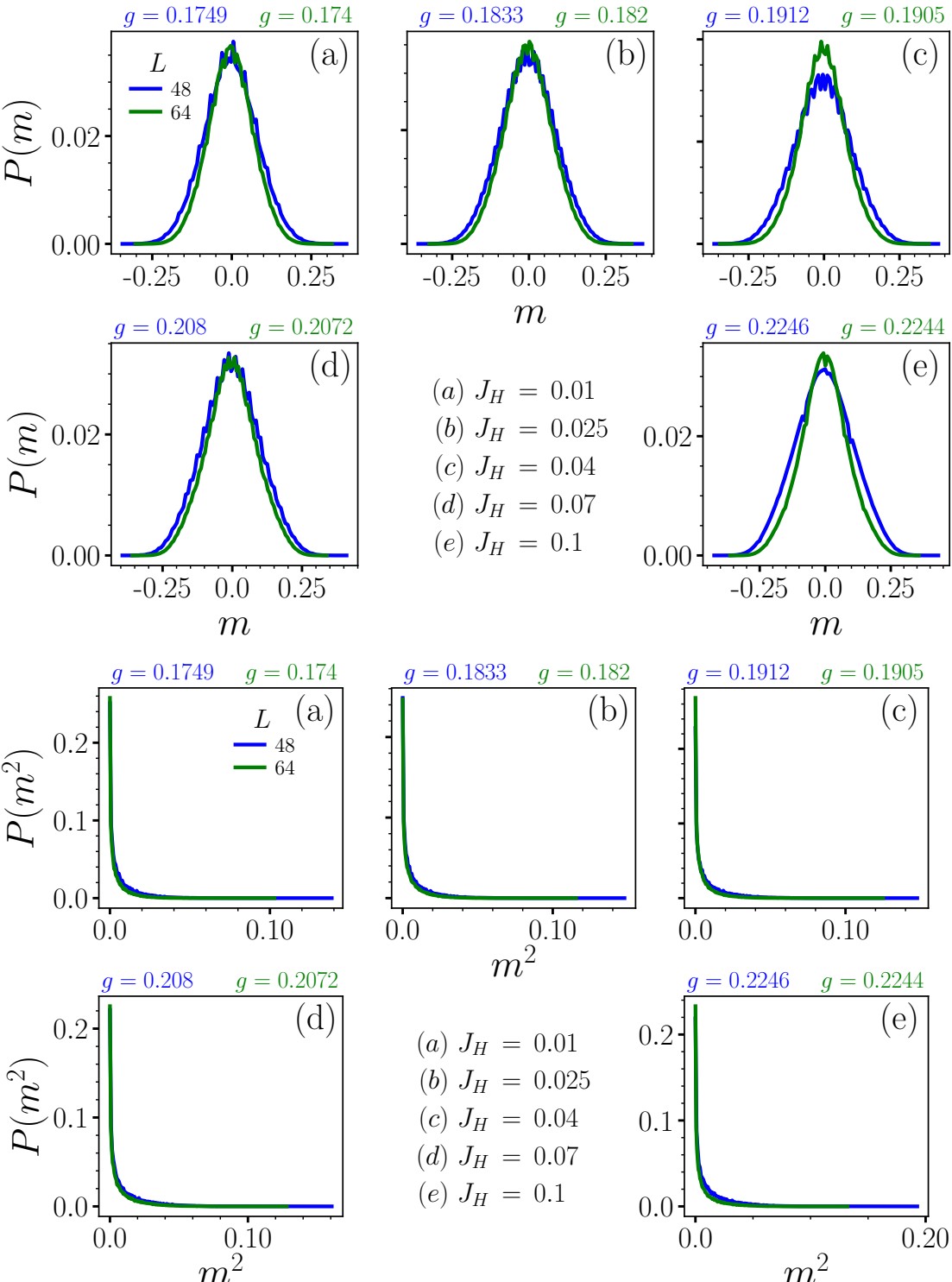

FIG. 6.    This figure expands on Fig. 2a of the main text to document the various staggered magnetization histogram data collected. $\beta = \frac{L}{4}$ throughout. The $g$ values above correspond to $\sim g_c(L)$.

## D. cVBS order parameter histograms

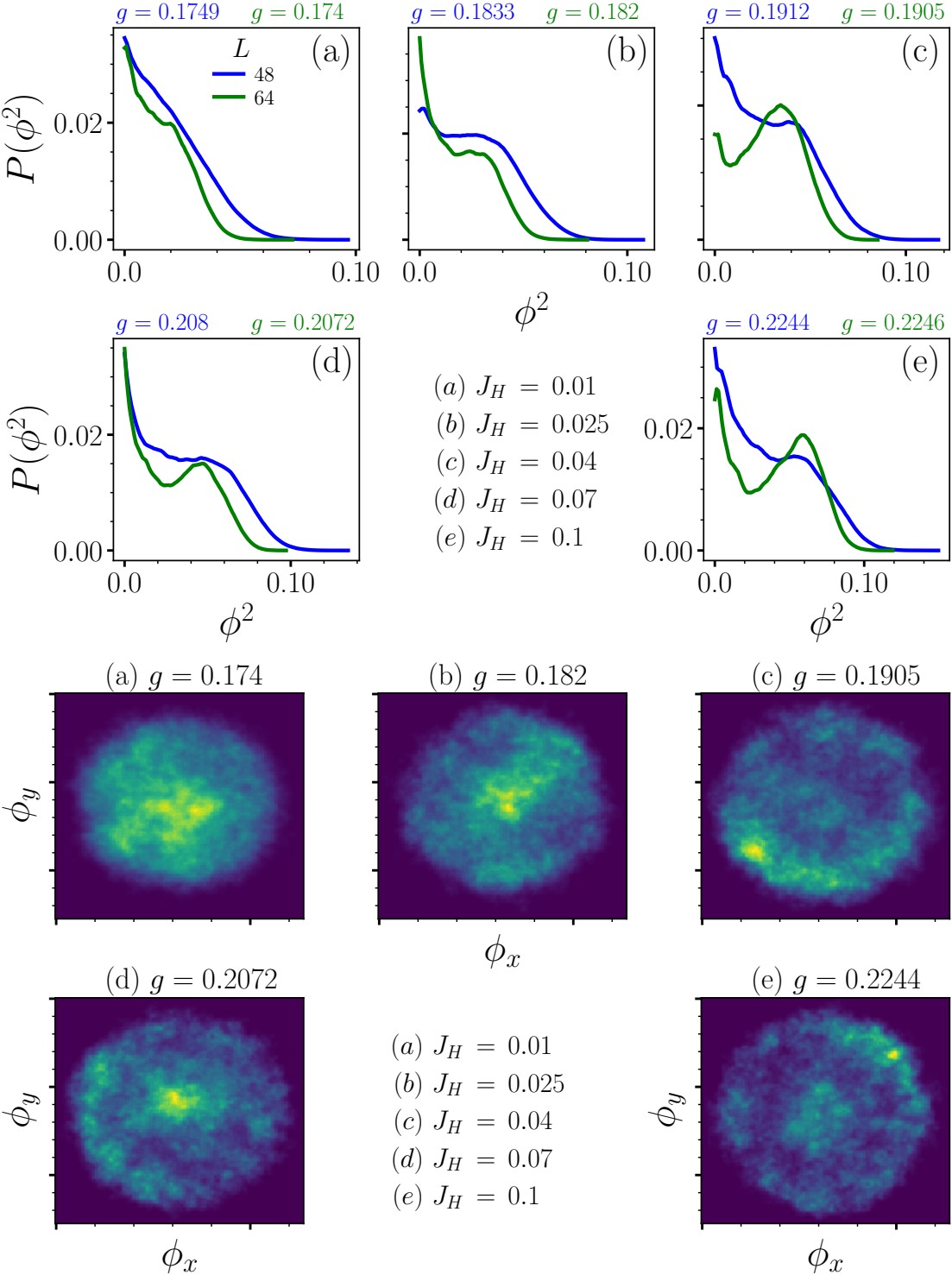

FIG. 7. This figure expands on Fig. 2b of the main text to document the various cVBS histogram data collected. $\beta = \frac{L}{4}$ throughout. $L = 64$ for cVBS $(\phi_x, \phi_y)$ heat maps. The $g$ values above correspond to $\sim g_c(L)$.

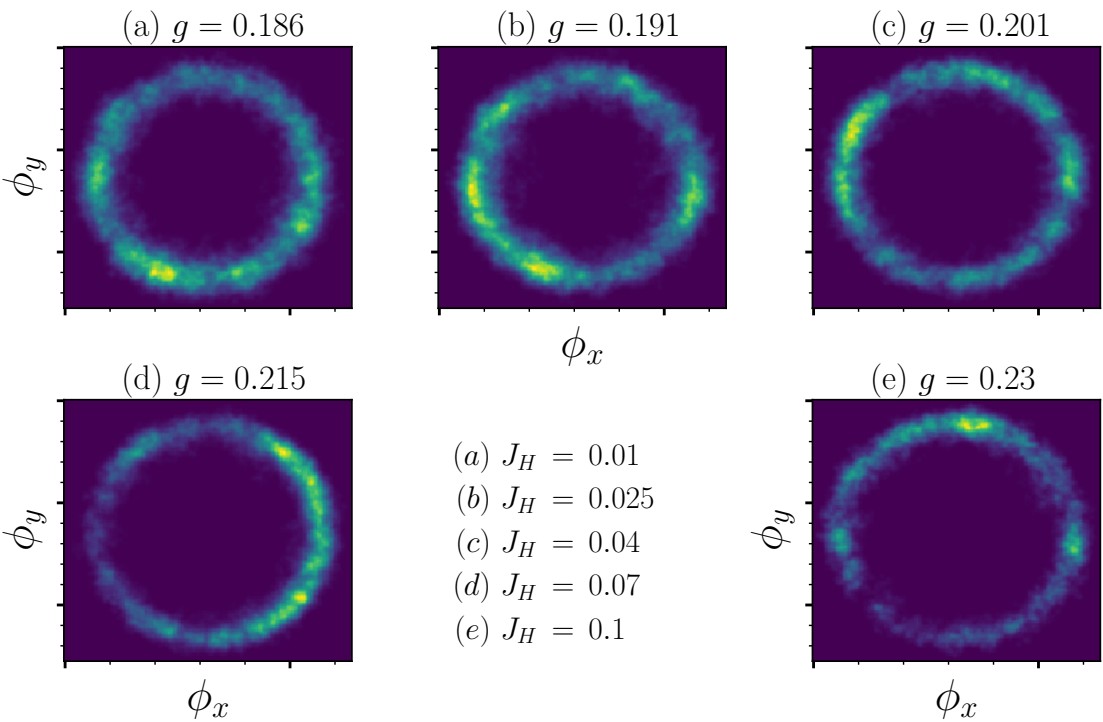

FIG. 8.   cVBS $(\phi_x, \phi_y)$ heat maps for $g$ slightly to the right of $g_C(L)$ on the VBS side to better highlight the "$U(1)$-symmetric" nature of these histograms near the phase transition. $\beta = \frac{L}{4}$ and $L = 64$ throughout.

### E.  Staggered magnetization time series

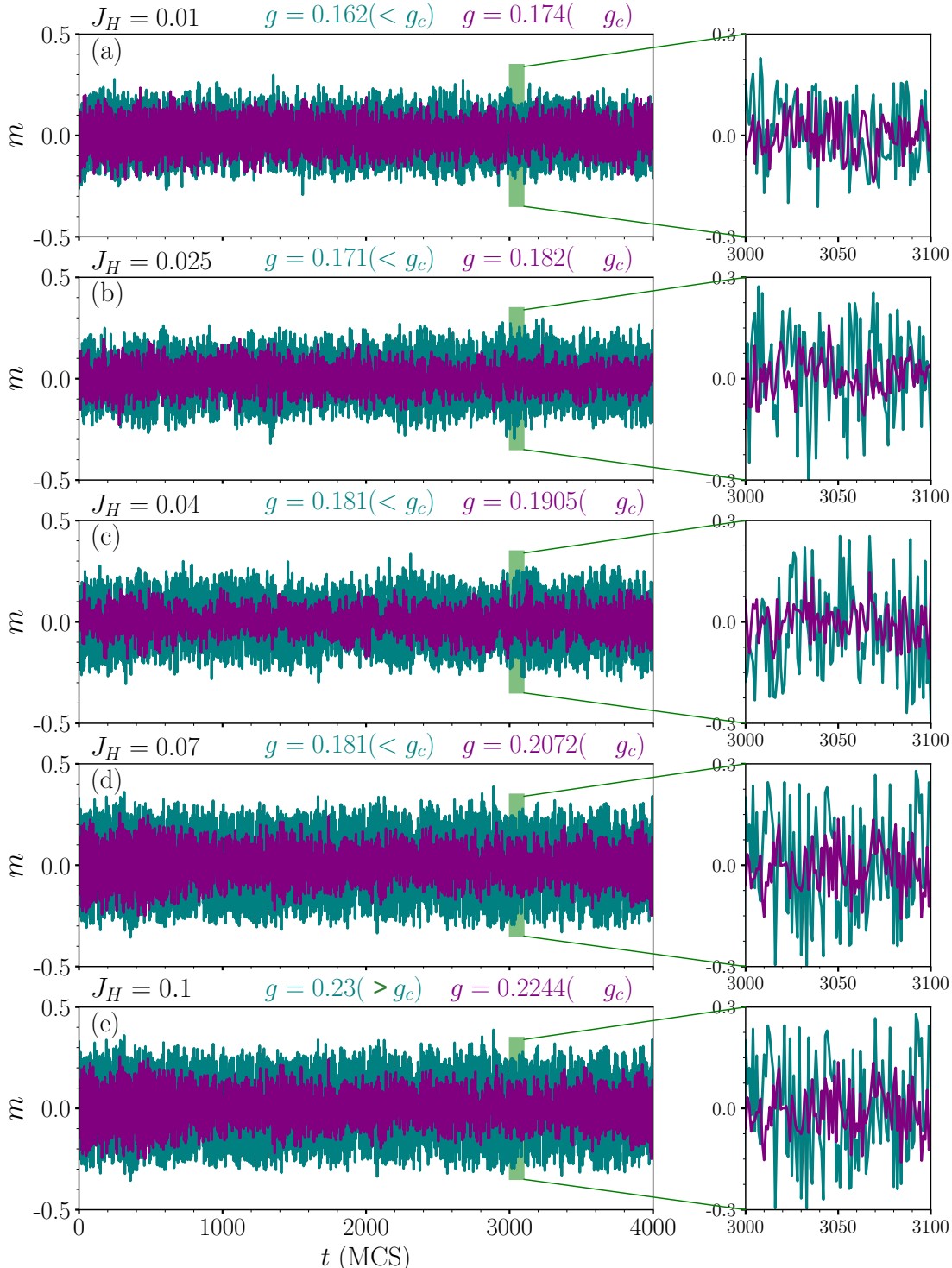

FIG. 9.  This figure expands on Fig. 2c to document the various time series data collected on staggered magnetization. $\beta = \frac{L}{4}$ and $L = 64$ throughout. The time series data are in register with those in Fig. 10.

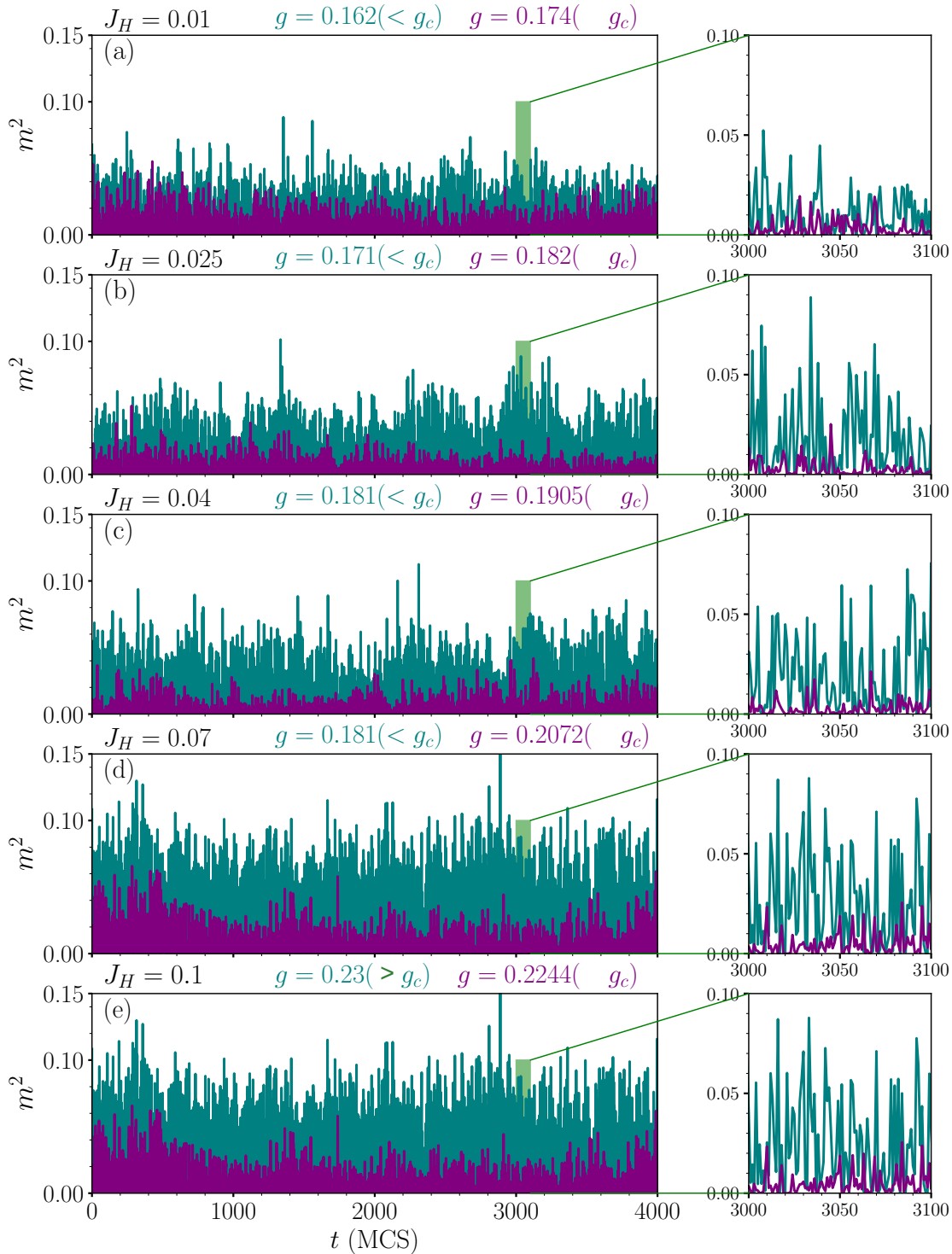

FIG. 10. This figure expands on Fig. 2c to document the various time series data collected on staggered magnetization. $\beta = \frac{L}{4}$ and $L = 64$ throughout. The time series data are in register with those in Fig. 9.

## F. cVBS order parameter time series

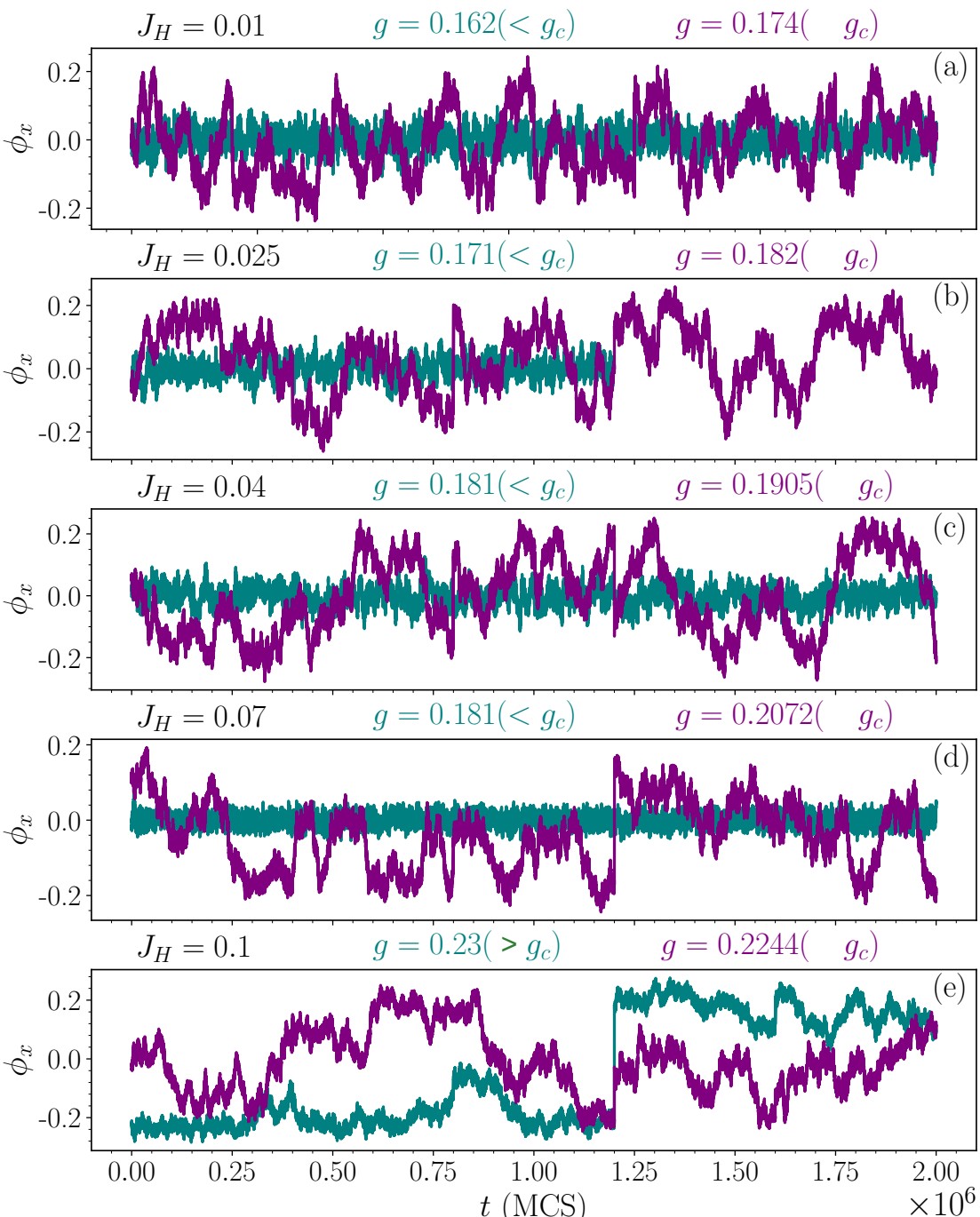

FIG. 11. This figure expands on Fig. 2 to document the various time series data collected on cVBS order parameter. $\beta = \frac{L}{4}$ and $L = 64$ throughout. The time series data are in register with those in Figs. 12,13.

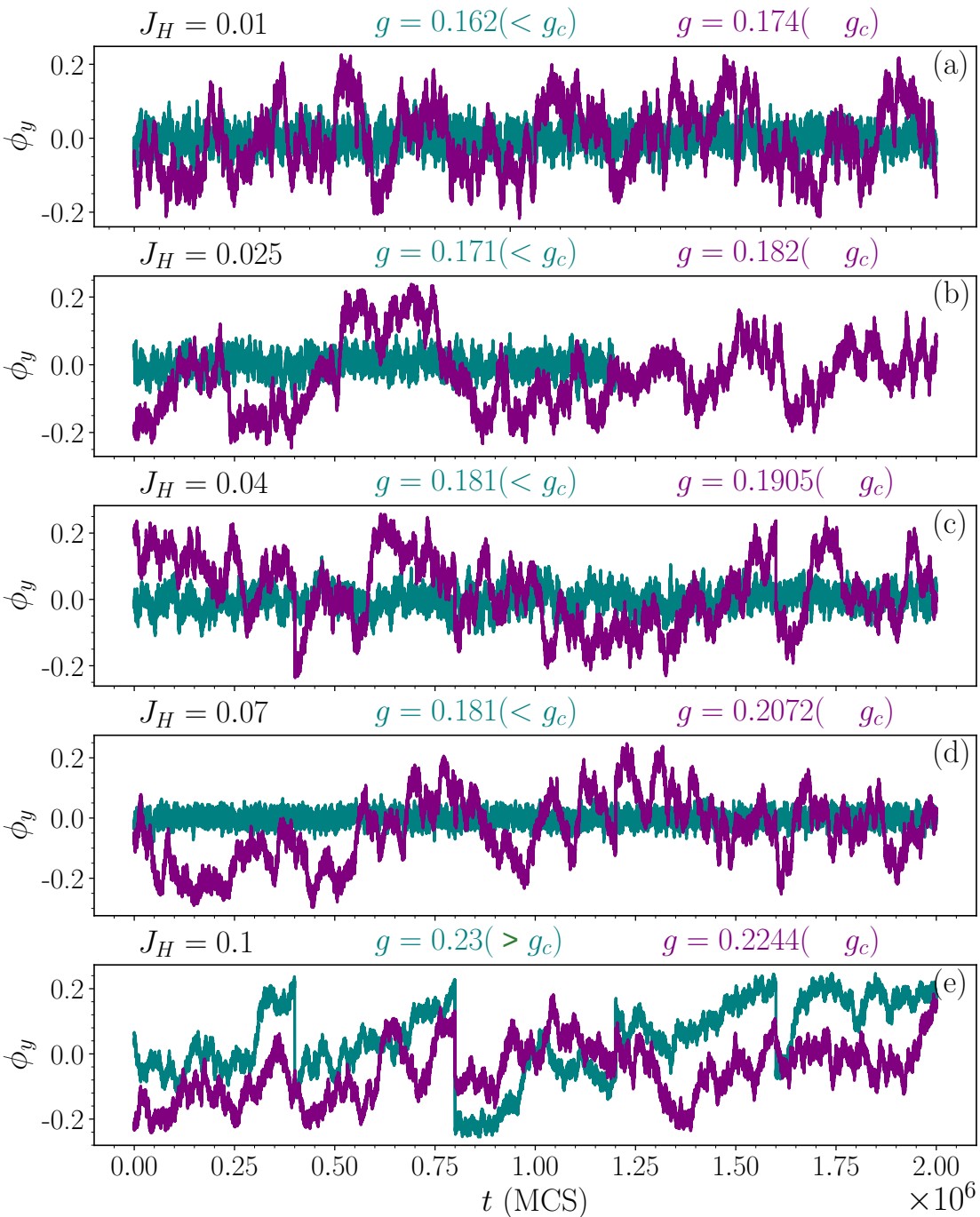

FIG. 12. This figure expands on Fig. 2 to document the various time series data collected on cVBS order parameter. $\beta = \frac{L}{4}$ and $L = 64$ throughout. The time series data are in register with those in Figs. 11,13.

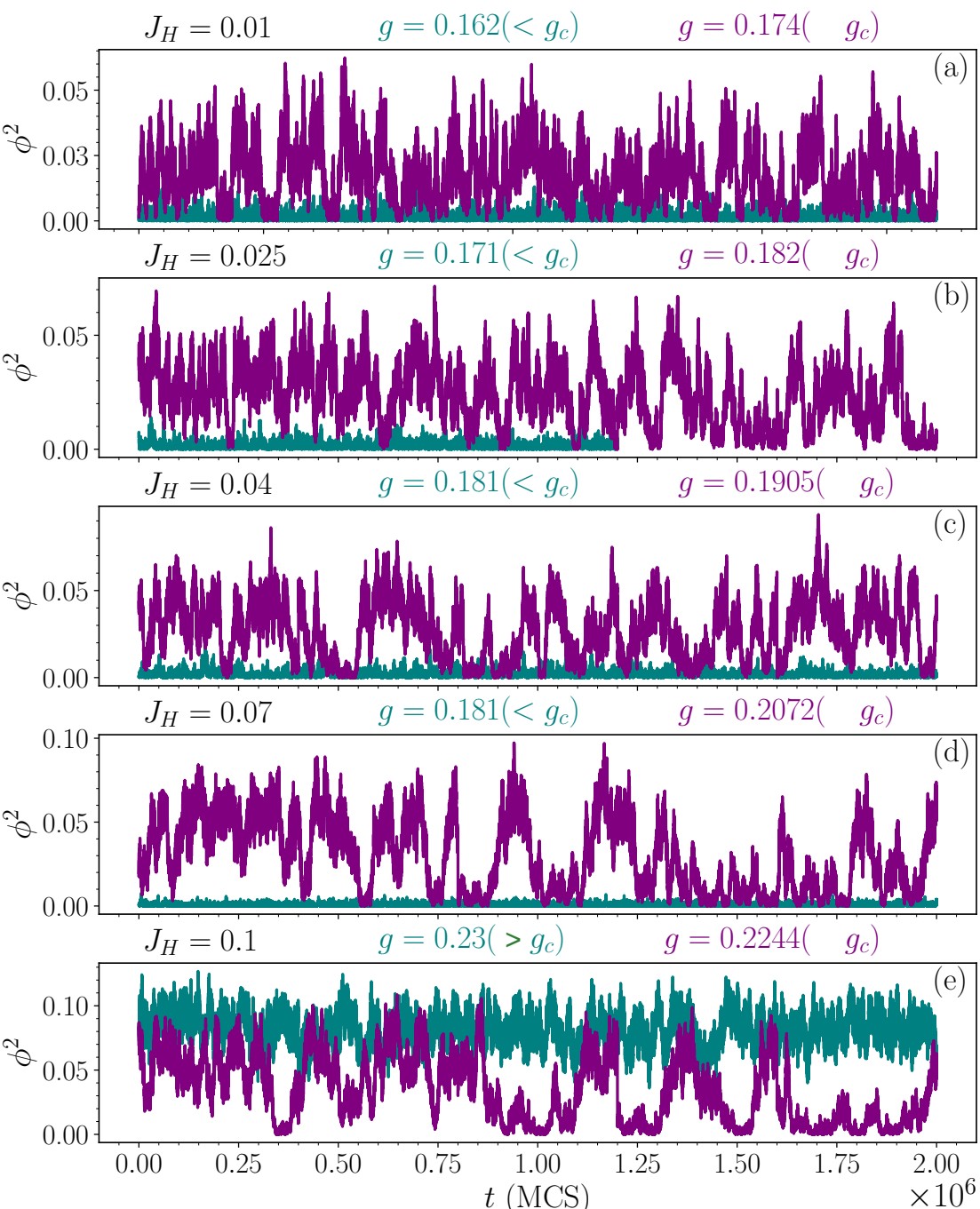

FIG. 13. This figure expands on Fig. 2 to document the various time series data collected on cVBS order parameter. $\beta = \frac{L}{4}$ and $L = 64$ throughout. The time series data are in register with those in Figs. 11,12.

## G.   Scaling collapse of correlation ratios and order parameters

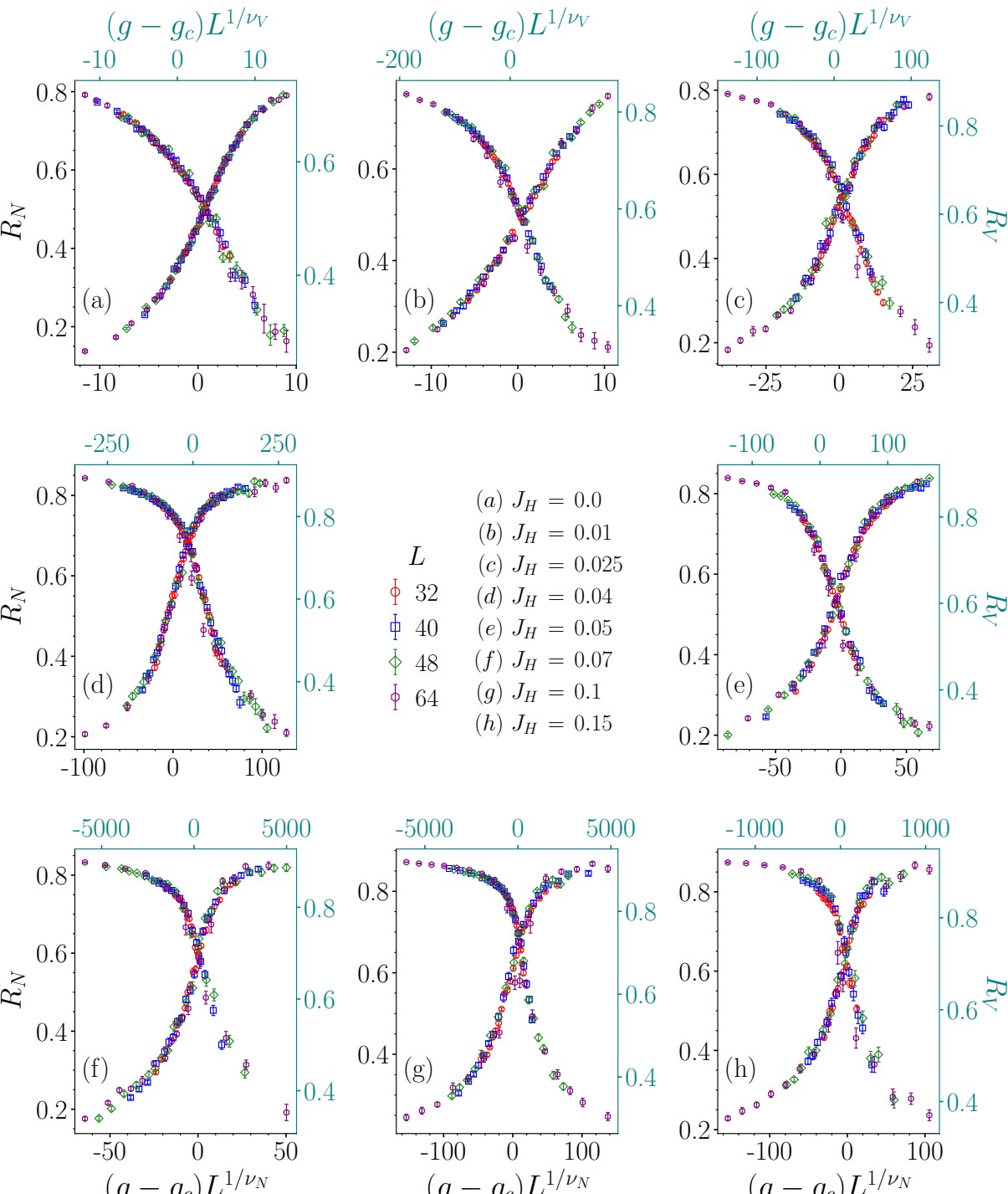

FIG. 14.    This figure expands on Fig. 3a,b of the main text to document all the scaling collapse analysis performed on correlation ratios. $\beta = \frac{L}{4}$ throughout.

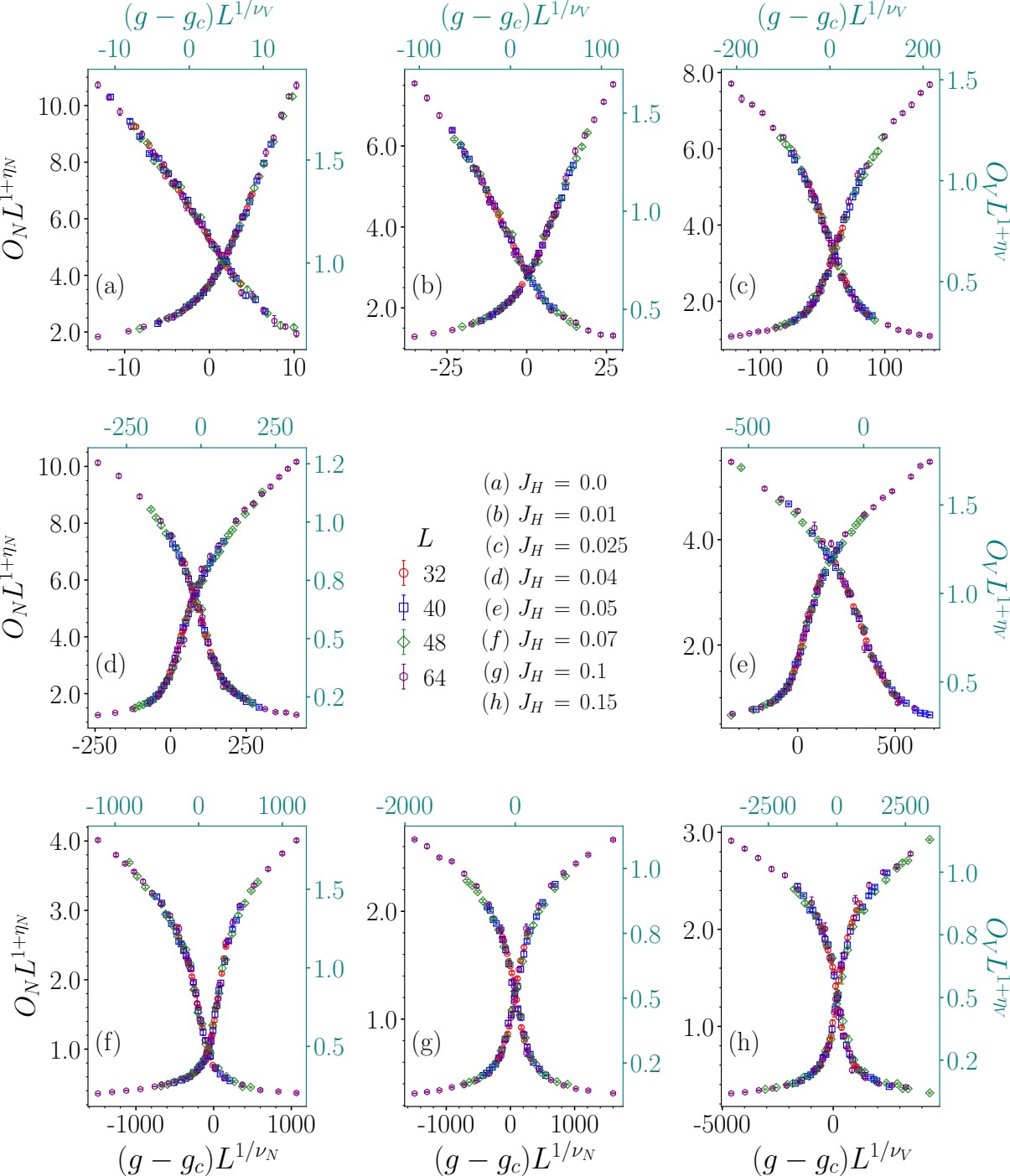

FIG. 15. This figure expands on Fig. 3c,d of the main text to document all the scaling collapse analysis performed on correlation ratios. $\beta = \frac{L}{4}$ throughout. $\beta = \frac{L}{4}$ throughout.

## H.   Binder Ratios

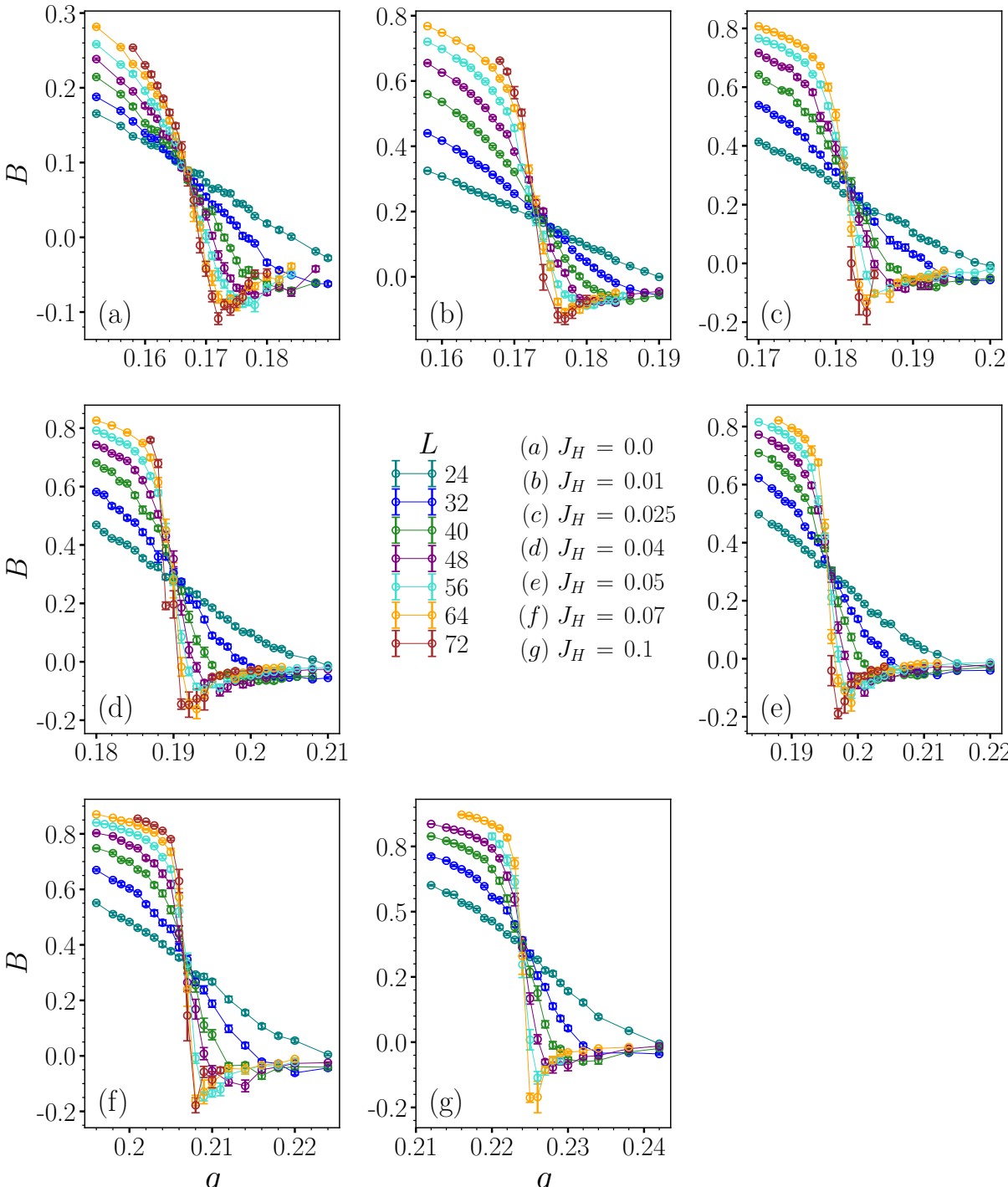

FIG. 16.   This figure expands on Fig. 4 of the main text to document all the Binder ratio data collected. $\beta = \frac{L}{4}$ throughout.

# III. TABLES

## A. Critcal exponents and critical point estimates

TABLE I. Critical exponents $(\nu_N, \nu_V)$ and critical point estimates $(g_{cN}, g_{cV})$ as obtained after performing a scaling collapse on the Néel and cVBS correlation ratios $(R_N$ and $R_B)$. System sizes used for collapses are in the range of $L = 24, 32, 40, 48, 64$. $\beta = \frac{L}{4}$ throughout. Neither $(\nu_N, \nu_V)$ nor $(g_{cN}, g_{cV})$ were set equal.

| $J_H$ | $\nu_N$ | $\nu_V$ | $g_{cN}$ | $g_{cV}$ | $\chi^2_N$ | $\chi^2_V$ |
|---|---|---|---|---|---|---|
| 0.0 | 0.49(5) | 0.63(1) | 0.168(1) | 0.167(1) | 0.9-1.56 | 1.14-1.81 |
| 0.01 | 0.41(2) | 0.56(4) | 0.175(1) | 0.171(1) | 1.31-2.27 | 1.32-1.84 |
| 0.025 | 0.40(3) | 0.51(3) | 0.182(1) | 0.18(1) | 0.97-2.43 | 1.15-1.61 |
| 0.04 | 0.37(4) | 0.49(1) | 0.191(1) | 0.188(1) | 1.12-2.64 | 1.14-1.86 |
| 0.05 | 0.40(3) | 0.46(3) | 0.196(1) | 0.195(1) | 0.91-1.78 | 1.15-1.84 |
| 0.07 | 0.34(5) | 0.48(5) | 0.207(1) | 0.206(1) | 1.52-2.54 | 1.04-1.77 |
| 0.1 | 0.31(1) | 0.40(1) | 0.225(1) | 0.223(1) | 1.55-2.27 | 1.03-2.26 |
| 0.15 | 0.32(4) | 0.44(2) | 0.254(1) | 0.252(1) | 1.37-3.39 | 0.74-1.91 |

TABLE II. Critical exponents $(\eta_V, \eta_N, \nu_N, \nu_V)$ and critical point estimates $(g_{cN}, g_{cV})$ as obtained after performing a scaling collapse of the Néel and cVBS order parameters $(O_N, O_B)$. System sizes used for collapses are in the range of $L = 24, 32, 40, 48, 64$. $\beta = \frac{L}{4}$ throughout. $(\nu_N, \nu_V)$ were not set equal to each other, while $g_{cN}$ and $g_{cV}$ values were fixed while performing the scaling collapse to the values obtained from the scaling collapse of correlation ratios as shown the preceding table.

| $J_H$ | $\nu_N$ | $\nu_V$ | $\eta_N$ | $\eta_V$ | $g_{cN}$ | $g_{cV}$ | $\chi^2_N$ | $\chi^2_V$ |
|---|---|---|---|---|---|---|---|---|
| 0.0 | 0.53(3) | 0.63(1) | 0.41(1) | 0.50(2) | 0.168 | 0.167 | 1.01-1.71 | 1.67-2.53 |
| 0.01 | 0.45(3) | 0.54(3) | 0.26(2) | 0.46(2) | 0.174 | 0.171 | 1.45-1.66 | 1.74-2.29 |
| 0.025 | 0.43(3) | 0.46(4) | 0.21(3) | 0.33(1) | 0.182 | 0.180 | 1.04-1.55 | 0.86-1.37 |
| 0.04 | 0.40(2) | 0.43(5) | 0.24(2) | 0.23(2) | 0.191 | 0.189 | 1.99-2.28 | 1.49-2.77 |
| 0.05 | 0.39(4) | 0.38(5) | 0.16(3) | 0.17(3) | 0.196 | 0.195 | 1.28-1.54 | 0.98-1.9 |
| 0.07 | 0.38(2) | 0.36(2) | 0.05(1) | 0.12(3) | 0.207 | 0.206 | 1.67-2.35 | 2.2-2.68 |
| 0.1 | 0.33(2) | 0.41(2) | 0.12(2) | 0.28(7) | 0.225 | 0.223 | 2.48-2.81 | 2.55-2.86 |
| 0.15 | ?? | ?? | ?? | ?? | 0.254 | 0.252 | ?? | ?? |

TABLE III. Critical exponents $(\eta_V, \eta_N, \nu_N, \nu_V)$ and critical point estimates $(g_{cN}, g_{cV})$ as obtained after performing a scaling collapse of Néel and cVBS order parameters $(O_N, O_B)$ for eight sets of the Heisenberg strength, $J_H = 0., 0.01, 0.025, 0.04, 0.05, 0.07, 0.1, 0.15$. System sizes used for collapses are in the range of $L = 24, 32, 40, 48, 64$. $\beta = \frac{L}{4}$ throughout. Neither $(\nu_N, \nu_V)$ nor $(g_{cN}, g_{cV})$ were set equal. This table was presented in the main text as well, and the estimates of various fitting parameters below are corroborated well by the estimates from the preceding tables.

| $J_H$ | $\nu_N$ | $\nu_V$ | $\eta_N$ | $\eta_V$ | $g_{cN}$ | $g_{cV}$ | $\chi^2_N$ | $\chi^2_V$ |
|---|---|---|---|---|---|---|---|---|
| 0.0 | 0.53(3) | 0.63(1) | 0.44(5) | 0.49(2) | 0.168(1) | 0.167(1) | 1.08-1.68 | 1.69-2.46 |
| 0.01 | 0.45(2) | 0.54(3) | 0.23(3) | 0.42(4) | 0.174(1) | 0.171(1) | 1.19-1.63 | 1.38-1.73 |
| 0.025 | 0.43(3) | 0.46(4) | 0.15(9) | 0.38(2) | 0.182(1) | 0.180(1) | 0.75-1.46 | 0.8-1.4 |
| 0.04 | 0.40(2) | 0.43(5) | 0.13(7) | 0.30(8) | 0.19(1) | 0.189(1) | 1.06-1.67 | 1.09-1.5 |
| 0.05 | 0.39(4) | 0.38(5) | 0.20(9) | 0.29(6) | 0.196(1) | 0.195(1) | 0.87-1.31 | 0.87-1.96 |
| 0.07 | 0.38(2) | 0.39(3) | 0.10(4) | 0.10(4) | 0.207(1) | 0.206(1) | 1.52-2.54 | 1.04-1.77 |
| 0.1 | 0.35(4) | 0.35(3) | -0.03(5) | -0.03(2) | 0.224(1) | 0.224(1) | 1.24-3.28 | 0.99-1.97 |
| 0.15 | 0.33(2) | 0.33(1) | 0.00(8) | -0.12(8) | 0.253(1) | 0.253(1) | 1.42-1.79 | 1.15-1.63 |

## B. Benchmarking with Exact Diagonalization

TABLE IV. This benchmarking table shows the values of total energy ($E$), Néel observables ($O_N$, $R_N$) obtained by Exact diagonalization (ED) and SSE-QMC on a $2 \times 2$ square plaquette at $\beta = 10$.

| $(J_B, Q_B, J_H)$ | $E^{\mathrm{ED}}$ | $E^{\mathrm{SSE}}$ | $O_N^{\mathrm{ED}}$ | $O_N^{\mathrm{SSE}}$ | $R_N^{\mathrm{ED}}$ | $R_N^{\mathrm{SSE}}$ |
|---|---|---|---|---|---|---|
| (1.0,1.0,0.2) | -5.4524 | -5.452(5) | 1.6980 | 1.6980(2) | 0.35739 | 0.3573(9) |
| (0.9,0.4,0.3) | -3.6835 | -3.683(2) | 1.7386 | 1.738(5) | 0.36655 | 0.3665(4) |
| (0.6,0.5,0.5) | -3.8492 | -3.849(0) | 1.7849 | 1.785(2) | 0.37651 | 0.3765(3) |
| (0.3,0.8,0.2) | -3.4539 | -3.4539(6) | 1.7170 | 1.717(1) | 0.36172 | 0.3617(1) |
| (1.15,0.88,0.12) | -5.2108 | -5.210(7) | 1.6861 | 1.686(2) | 0.35463 | 0.3546(0) |

TABLE V. This benchmarking table shows the values of VBS observables ($O_V$, $R_V$, $O_B$, $R_B$) obtained by Exact diagonalization (ED) and SSE-QMC on a $2 \times 2$ square plaquette at $\beta = 10$.

| $(J_B, Q_B, J_H)$ | $O_V^{\mathrm{ED}}$ | $O_V^{\mathrm{SSE}}$ | $R_V^{\mathrm{ED}}$ | $R_V^{\mathrm{SSE}}$ | $O_B^{\mathrm{ED}}$ | $O_B^{\mathrm{SSE}}$ | $R_B^{\mathrm{ED}}$ | $R_B^{\mathrm{ED}}$ |
|---|---|---|---|---|---|---|---|---|
| (1.0,1.0,0.2) | 1.63654 | 1.6365(4) | 0.5 | 0.4999(5) | 1.49930 | 1.4993(3) | 0.5 | 0.5000(1) |
| (0.9,0.4,0.3) | 1.65070 | 1.6507(4) | 0.5 | 0.4999(8) | 1.49615 | 1.496(2) | 0.5 | 0.49999(8) |
| (0.6,0.5,0.5) | 1.66564 | 1.665(5) | 0.5 | 0.49999(7) | 1.48881 | 1.488(8) | 0.5 | 0.4999(9) |
| (0.3,0.8,0.2) | 1.64326 | 1.643(4) | 0.5 | 0.4999(9) | 1.49817 | 1.498(0) | 0.5 | 0.5000(3) |
| (1.15,0.88,0.12) | 1.63223 | 1.632(1) | 0.5 | 0.4999(5) | 1.49973 | 1.499(8) | 0.5 | 0.49999(3) |