# Peer review of "Deconfined pseudocriticality in a model spin-1 quantum antiferromagnet"

_SciPost Physics Core_

## Round 3 · Referee Report · Anonymous (Referee 1) · 2025-1-4

Strengths

  1. High quality numerics. The method (SSE QMC) is rigorous, the quality of the simulation data is excellent and careful finite-size scaling has been done to obtain estimates of properties in the thermodynamic limit.
  2. The microscopic model is chose judiciously to include the dominant competing interactions. This reflects the expertise of the team in this problem.
  3. The most important physical observables are calculated to a high accuracy and analysed thoroughly to draw the conclusion reached in the paper.

Weaknesses

  1. What is the motivation for choosing the different terms in the Hamiltonian? In particular, what is the rationale for keeping both J_H and J_B?
  2. Calculation of the Renyi entropy would have given further confirmation whether it truly is a 1st order transition.
  3. I believe "deconfined pseudo criticality" means "close to a DQC point". Some discussion (even if speculative) on what kind of interactions would drive the system towards DQC would have been very helpful.
  4. In general, the hamiltonian seems to be a trivial extension of Desai's PRL from 2019. What is new in this model and why's is that significant?
  5. Can the authors identify the nature of spinons that get deconfined at the proximate DQC point?

Report

The authors have presented numerical investigation into possible DQC in a spin-1 system on a square lattice. They have chosen an extension of the Hamiltonian used in Desai's previous work. Using high quality QMC simulations and careful finite size scaling analysis of most important physical observables, they establish that the transition from Neil to VBS is weakly first order and exhibits strong signatures of spinon deconfinement, which they (rightly) ascribe to the proximity to a DQC point.

The problem is of great current interest. However, despite its significant technical strengths, the work falls short of contributing strongly to moving the field forward. Addressing the weaknesses listed above will help making it a better .

Requested changes

  1. Justify the choice of the Hamiltonian.
  2. Calculate Renyi Entropy.
  3. Identify nature of piñons.
  4. Discuss possible routes to DQC.

Recommendation

Ask for major revision

---

## Round 3 · Referee Report · Anonymous (Referee 2) · 2025-1-12

Strengths

1) Study of deconfined quantum criticality (DQC) in the context of spin-1 systems 2) High quality numerical data

Weaknesses

1) Lack of motivation for this particular spin system 2) Interpretation of results in view of DQC (too) speculative 3) Further observables could provide more concrete picture for interpretation of results

Report

In this manuscript, the authors study a model candidate to feature deconfined quantum criticality (DQC) in a spin-1 system. In particular, they investigate the SU(3) JQ model in the presence of an additional SU(2) perturbation by means of unbiased quantum Monte Carlo simulations. Here, the authors show that a direct cVBS to Néel transition persists in the presence of weak SU(2) perturbations and further discuss the nature of the phase transition in view of the DQC behavior in the SU(3) case.

While the topic of DQC in the context of spin-1 systems is certainly interesting and the authors provide excellent data along with a thorough numerical analysis, in my opinion the manuscript in its current form is not (yet) suitable for publication in SciPost Physics Core. The draft currently lacks motivation for the investigation of this particular spin model. Further, some additional observables are necessary to assess the scenario of weakly first-order and pseudocriticality due to proximity to a DQC point as suggested in this draft.

Below are some remarks I kindly address to the authors which they may consider for their revision.

Requested changes

1) Given that what happens even in the pure SU(3) case is still under debate, it is not clear to me why the introduction of the additional SU(2) perturbation is beneficial. It would be very helpful to add some more motivation for this particular choice of Hamiltonian to the introduction.

2) I find the terminology of "deconfined pseudocriticality" used in the title and throughout the manuscript a bit confusing. To my understanding, this is based on the assumption of DQC as weakly first-order pseudocriticality and the additional assumption that introducing the SU(2) perturbation does not alter this scenario. In my opinion, the manuscript would benefit from introducing more concretely what the authors mean by this terminology and/or using a more specific title.

3) Regarding the issue of the nature of the phase transition the authors cite arXiv:2106.15462 (Ref. [75] in the manuscript), where Emidio et al. introduce a method to distinguish weakly first-order and continuous phase transitions which they applied to the spin-1/2 JQ model. Is this also usable for the spin-1 case discussed in this manuscript? This approach could shed some (more) light on the nature of the phase transition and complement the Binder ratio analysis provided here. In my opinion the authors should try this method or at least comment on it.

4) Regarding the measurement of the critical exponents shown in Table 1 and Figure 3: To my understanding the authors take $\nu_N \sim \nu_V$ as one criterion for the DQC like behavior in this model. For the case of $J_H=0$ and $J_H=0.01$, i.e. where the DQC like behavior is expected to be strongest, the exponents do not agree within their statistical accuracies. Why is that the case? The authors further state that "setting them equal does not lead to significant loss of collapse quality". What happens in this case with the anomalous exponents $\eta_N$ and $\eta_V$? Further, if setting the exponents equal does not alter the quality of the fit, this may raise doubts how reliable the exponent estimation is in these cases. Here, some additional discussion would be helpful.

Following are some minor questions/remarks the authors may consider for their revision:

5) For readability it would be beneficial to move the definition of the correlation ratios $R$ to the main text rather than a footnote. 6) Regarding the formulation of scenario 1) on page 2: Isn't it more fitting to state that the criticality may become "strongly" first order? 7) Regarding the formulation of scenario 2) on page 2: I think there is a typo here "there is a regime where the ... obtains" (obtains --> persists) 8) The Binder parameter $B$ on page 3. could be introduced more clearly earlier as $B=...$ (since $B$ wasn't defined otherwise).

Recommendation

Ask for major revision

---

## Editorial Decision

awaiting_resubmission